# Observation of negative capacitance in antiferroelectric PbZrO₃ Films

Leilei Qiao[1], Cheng Song [1✉], Yiming Sun[1], Muhammad Umer Fayaz [1], Tianqi Lu[2], Siqi Yin[1], Chong Chen[1], Huiping Xu[1], Tian-Ling Ren [2] & Feng Pan[1✉]

Negative capacitance effect in ferroelectric materials provides a solution to the energy dissipation problem induced by Boltzmann distribution of electrons in conventional electronics. Here, we discover that besides ferroelectrics, the antiferroelectrics based on Landau switches also have intrinsic negative capacitance effect. We report both the static and transient negative capacitance effect in antiferroelectric PbZrO₃ films and reveal its possible physical origin. The capacitance of the capacitor of the PbZrO₃ and paraelectric heterostructure is demonstrated to be larger than that of the isolated paraelectric capacitor at room temperature, indicating the existence of the static negative capacitance. The opposite variation trends of the voltage and charge transients in a circuit of the PbZrO₃ capacitor in series with an external resistor demonstrate the existence of transient negative capacitance effect. Strikingly, four negative capacitance effects are observed in the antiferroelectric system during one cycle scan of voltage pulses, different from the ferroelectric counterpart with two negative capacitance effects. The polarization vector mapping, electric field and free energy analysis reveal the rich local regions of negative capacitance effect with the negative $dP/dE$ and $(\delta^2 G)/(\delta D^2)$, producing stronger negative capacitance effect. The observation of negative capacitance effect in antiferroelectric films significantly extends the range of its potential application and reduces the power dissipation further.

[1] Key Laboratory of Advanced Materials (MOE), School of Materials Science and Engineering, Beijing Innovation Center for Future Chip (ICFC), Tsinghua University, Beijing, China. [2] Institute of Microelectronics & Beijing National Research Center for Information Science and Technology (BNRist), Tsinghua University, Beijing, China. ✉email: songcheng@mail.tsinghua.edu.cn; panf@mail.tsinghua.edu.cn

The Boltzmann tyranny poses great challenges to the low-power dissipation and the device dimension scale-down in conventional electronics[1]. The negative capacitance (NC) effect in ferroelectrics was proposed to solve this problem, by passively boosting the gate voltage[2] and proved effective experimentally[3,4]. The NC effect are usually classified into the static NC effect and the transient NC effect[5]. The static NC effect can be applied in the transistor due to the hysteresis-free character, and can be stabilized by adding a positive capacitance in series[6–8]. This series circuit can drive the device from the voltage-controlled operational mode into charge-controlled one, which would reverse the polarization and voltage curve from the multivalued S shape to the single-valued N shape. The exploration into the mechanism of the static NC effect has developed from the initial monodomain Landau theory to the mechanism based on domains[5]. Different from the static NC effect, the transient NC effect is always accompanied by hysteresis and occurs during the polarization switching, suggesting this process is transient and not stable. Reverse domain nucleation and accelerated growth during the switching can induce the switching instability, making the temporal evolution of the charge, and voltage follows a negative slope[9,10]. By applying a voltage pulse on the NC capacitor in series with an external resistor ($R$–$C$ circuit), the voltage drop across the ferroelectric is observed in the limited time range with charge $Q$ increasing[10,11]. Given that the transient character of the transient NC effect, it may not be applied for the low-power logic transistor.

The observation of NC effect is focused on the ferroelectrics up to now[11,12]. Besides ferroelectrics, the antiferroelectrics based on Landau switches may have NC effect. For the ideal monodomain antiferroelectrics, the second derivative between the free energy $G$ and the charge $Q$ of the free energy curves could lead to the NC effect[13,14]. While the realistic structure in antiferroelectrics tends to be multidomain structures due to the competition between the energies[15], and the multidomain structures have been observed in our samples (Supplementary Fig. S1). The sublattice structures in antiferroelectrics further complicate the system and would enrich the physical origin. In this work, we report a proof-of-concept demonstration of static NC effect by capacitance enhancement in the paraelectric–antiferroelectric heterostructure capacitor. Meanwhile, the direct measurements of the transient NC effect are realized by constructing a simple circuit of an antiferroelectric capacitor in series with an external resistor by monitoring the voltage and charge transients. The polarization vector mapping, the internal electric field and the free energy analysis reveal that rich NC effect local regions emerge in antiferroelectric film.

The observation of NC effect is focused on the ferroelectrics up to now[11,12]. Besides ferroelectrics, the antiferroelectrics based on Landau switches may have NC effect. For the ideal monodomain antiferroelectrics, the second derivative between the free energy $G$ and the charge $Q$ of the free energy curves could lead to the NC effect[13,14]. While the realistic structure in antiferroelectrics tends to be multidomain structures due to the competition between the energies[15], and the multidomain structures have been observed in our samples (Supplementary Fig. S1). The sublattice structures in antiferroelectrics further complicate the system and would enrich the physical origin. In this work, we report a proof-of-concept demonstration of static NC effect by capacitance enhancement in the paraelectric–antiferroelectric heterostructure capacitor. Meanwhile, the direct measurements of the transient NC effect are realized by constructing a simple circuit of an antiferroelectric capacitor in series, with an external resistor by monitoring the voltage and charge transients. The polarization vector mapping, the internal electric field, and the free energy analysis reveal that rich NC effect local regions emerge in antiferroelectric film.

## Results

**Capacitance enhancement induced by NC effect in the antiferroelectric capacitor.** PbZrO$_3$ (PZO) films are used to explore the NC effect in antiferroelectric films for its characteristic antiferroelectric performance. The X-ray diffraction (XRD) measurement results of PZO films are shown in Fig. 1a. The peaks marked by the red pentagons indicate the pure orthorhombic phase, which suggests the antiferroelectric characteristics of crystallizable PZO films[16]. In the meanwhile, the large saturation polarization strength of 50 $\mu$C cm$^{-2}$ and weak remanent polarization of the PZO film in Fig. 1b further demonstrate the antiferroelectricity. The weak remanent polarization most likely originates from the unique dipolar arrangement modes in the PZO film[17], which would be discussed later. The antiferroelectric phase is transformed to the ferroelectric phase when the applied electric filed is over ~0.45 MV cm$^{-1}$, and the ferroelectrics are recovered to antiferroelectrics when the electric field is lower than ~0.2 MV cm$^{-1}$.

Figure 1c displays the small signal capacitance–voltage ($C$–$V$) response of the crystallizable PZO (50 nm) and amorphous PZO (a-PZO, 50 nm) capacitors with a frequency of 100 KHz under a direct bias from −3 to 3 V at room temperature. The capacitance of the a-PZO film keeps almost constant of 3 pF with negligible hysteresis under the voltage range fro −3 to 3 V, revealing its paraelectric characteristics. While the $C$–$V$ curve of the crystallizable PZO capacitor displays characteristic antiferroelectricity with double-butterfly shape and four peaks through the whole voltage range, demonstrating the antiferroelectric performance further. The antiferroelectric PZO holds higher capacitance density and more significant hysteresis.

For two positive capacitors in series, the overall capacitance $C$ must be positive due to the thermodynamic stability and is always smaller than that of any part, which is decided by the general computational formula of the series capacitance: $1/C = 1/C_1 + 1/C_2$ ($C$ represents the overall capacitance; $C_1$ and $C_2$ are two capacitors that are series in the circuit). But the relationship is also true if one of the local effective capacitances, for example, $C_1$ is negative, so long as $C$ is still positive ($|C_1| > |C_2|$), leading to the result that the overall capacitance is higher than $C_2$. This capacitance enhancement effect has been widely adopted to observe the static NC effect in ferroelectric heterostructure and superlattice[6,18,19]. To demonstrate the static NC effect in antiferroelectric PZO films, two capacitors Pt(10 nm)/amorphous HfO$_2$ (a-HfO$_2$, 20 nm)/La$_{2/3}$Sr$_{1/3}$MnO$_3$(LSMO, 20 nm), Pt(10 nm)/a-HfO$_2$(20 nm)/PZO(50 nm)/LSMO(20 nm) were designed to compare the values of their capacitances. For a comparison, Pt(10 nm)/a-HfO$_2$(20 nm)/LSMO(20 nm) and Pt(10 nm)/a-HfO$_2$(20 nm)/a-PZO (50 nm)/LSMO(20 nm) capacitors were fabricated to verify the validity of the capacitance enhancement. The series circuit of the paraelectric and antiferroelectric capacitors enables the charge-driven operational mode which can reverse the double-S shape $P$–$E$ loop into the single-valued double-N shape curve, making NC effect accessible. Similar with the a-PZO films, the $C$–$V$ curves of Pt/a-HfO$_2$/LSMO and Pt/a-HfO$_2$/a-PZO/LSMO capacitors in Fig. 1d are both straight lines with the capacitance of 5.4 pF and 2.0 pF, respectively. The $C$–$V$ curve of Pt/a-HfO$_2$/PZO/LSMO capacitor does not show double-butterfly shape because the voltage range is narrow so that it cannot show the complete curve in order to avoid the dielectric breakdown induced by the overlarge voltage. For Pt/a-HfO$_2$/a-PZO/LSMO capacitor, the capacitance is clearly lower than that of Pt/a-HfO$_2$/LSMO capacitor, which is consistent with the formula of $1/C = 1/C_1 + 1/C_2$. Remarkably, the capacitance of Pt/a-HfO$_2$/PZO/LSMO capacitor is found to be larger than the constituent a-HfO$_2$ capacitance over the entire voltage. This capacitance enhancement effect means that the antiferroelectric PZO film contributes the NC effect to the heterostructure.

The effective permittivity of 50 nm PZO film calculated from the $C$–$V$ loop is 61.33 in our work, which is lower than that of

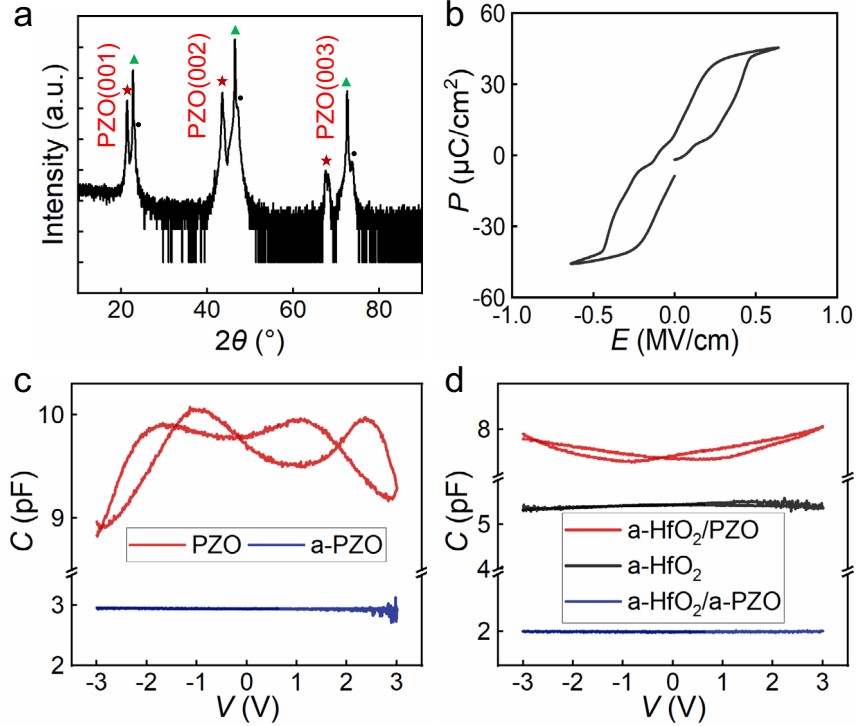

**Fig. 1 Capacitance enhancement induced by the static NC effect at room temperature. a** XRD data of the PZO film with the orthorhombic phase deposited on the LSMO bottom electrode with STO substrate. The peaks marked by green triangles represent the STO phase and the peaks marked by black dots represent the LSMO phase. The red stars represent the (00$l$) orientation in pseudocubic index of the PZO films. **b** Antiferroelectric polarization $P$ as a function of the electric field $E$ ($P$–$E$). **c** Small signal capacitance–voltage ($C$–$V$) characteristics measured at 100 kHz for the Pt/PZO/LSMO capacitor and Pt/a-PZO/LSMO capacitor. **d** Comparison of $C$–$V$ characteristics of Pt/a-HfO$_2$/PZO/LSMO, Pt/a-HfO$_2$/a-PZO/LSMO, and Pt/a-HfO$_2$/LSMO capacitors.

bulk PZO ceramics with 220 at the frequency of 100 KHz[20]. The relative permittivity of 20 nm a-HfO$_2$ film is calculated as 13.67 at the same frequency, which is comparable with the reported value. The permittivity is closely related to the crystal structure, growth quality, film thickness, measurement frequency, temperature, and so on[20]. Thus the difference of dielectric permittivity most likely results from the material system and the film quality. In order to exclude the influence of the Maxwell–Wagner effect, which can also lead to an enhancement of the effective permittivity at low frequency, the frequency dependance experiments are performed. The capacitance of another a-HfO$_2$/PZO sample is indeed higher than that of corresponding a-HfO$_2$ sample through the whole frequency range from 1 KHz to 5 MHz (Supplementary Fig. S2). This effect origins from the accumulation of charges at the interface of heterostructures or superlattices, and is related closely with the number of heterojunction interfaces and the measurement frequency[21]. Noted that the single interface in our structures and capacitance enhancement at high frequency, the capacitance enhancement effect originates from the NC effect of the antiferroelectric film, rather than the Maxwell–Wagner effect.

Room-temperature demonstration of the NC effect is an important step to put antiferroelectric materials into practical application, such as transistors to lower the subthreshold swing below the intrinsic thermodynamic limit of 60 mV per decade[6,7,22,23], and thereby improving the energy efficiency.

**Transient NC effect in antiferroelectrics under voltage pulses.** For nonlinear dielectrics, such as ferroelectric and antiferroelectric materials, the capacitance can be expressed by $C = \mathrm{d}Q/\mathrm{d}V$, different from $C = Q/V$ for linear capacitor. Accordingly, NC effect can take place in the region where the voltage and charge across the capacitor have opposite variation tendency as a

function of time, meaning that an increase (decrease) of the voltage can lead to the decrease (increase) of the charge[10]. But in the $C$–$V$ response measurements, the measurement is relatively slow, compared with the switching of the ferroelectric and antiferroelectric films, so that the capacitance keeps positive during the whole measurements. To detect the transient NC directly, a proper voltage pulse is applied across a series combination of an antiferroelectric capacitor and an external resistor $R$, where the resistor is used to slow down the delivery of screening charge. This method is proved to be effective to slow down the switching speed of polarization reversal so that the NC effect can be captured in ferroelectric capacitors-based $R$–$C$ circuit[10,24,25].

Figure 2a shows the schematic diagram of the experimental setup. The antiferroelectric capacitor Pt/PZO/LSMO is in series with the resistor $R$ of 3 KΩ. The source voltage $V_s$ is an a.c. pulse as $-5 \rightarrow +5 \rightarrow -5$ V. The voltage across the antiferroelectric capacitor is calculated by $V_{\mathrm{AFE}} = V_s - IR$, also shown in Fig. 2b. The current $I$ through the series circuit is directly detected, as illustrated in Fig. 2c, and the total charge across the whole circuit is calculated by $Q(t) = \int_0^t I(t)\mathrm{d}t$, as presented in Fig. 2d. It is worth noting that the charge $Q$ here is not the charge across the antiferroelectric capacitor, because the parasitic capacitance $C_p$ in the circuit also contributes to the whole charge. The charge across the antiferroelectric capacitor $Q_{\mathrm{AFE}}$ can be calculated from $Q$: $Q_{\mathrm{AFE}}(t) = Q_{\mathrm{AFE}}(t=0) + Q(t) - C_p V_{\mathrm{AFE}}(t)$. Now, we use $Q$ to replace $Q_{\mathrm{AFE}}$ because $C_p$ is usually so small that can be neglected. We note in Fig. 2b that in the shadow area of region I, where just after the PZO film goes through the transition from antiferroelectric phase to ferroelectric phase, which is magnified in Fig. 2e, the voltage across the antiferroelectric capacitor begins to decrease after it increases to a high point. In the same time region, the current is positive and the charge increases, as displayed in Fig. 2c,

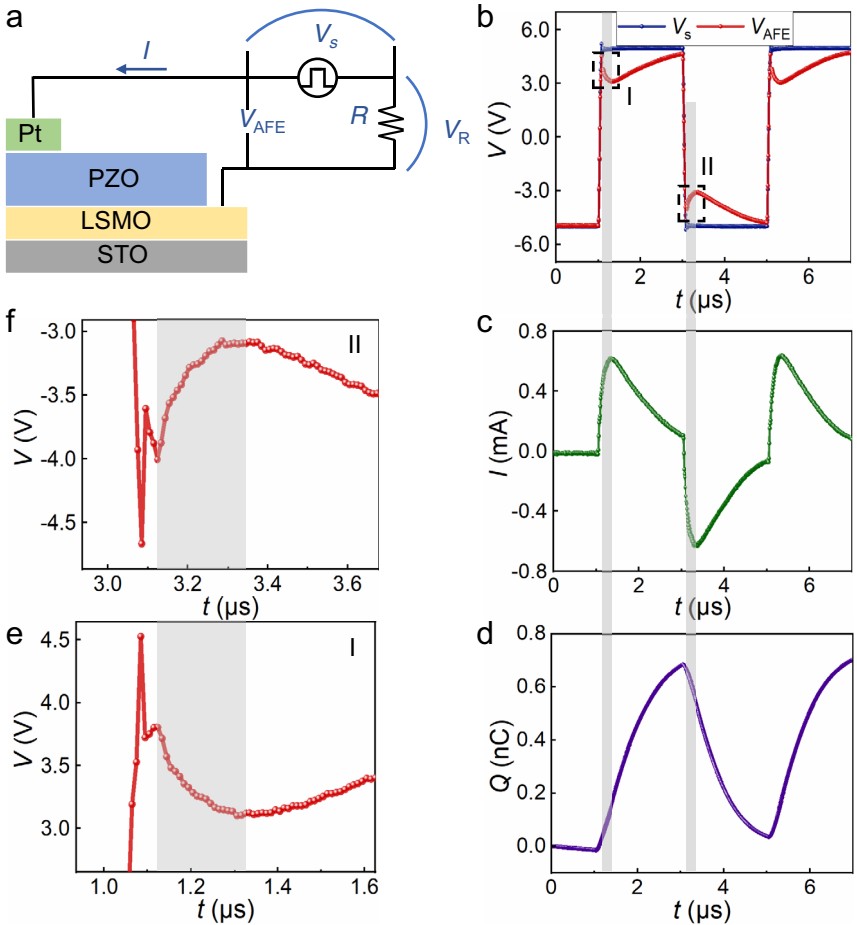

**Fig. 2 Transient response of an antiferroelectric capacitor. a** Schematic diagram of the experimental setup. Transients corresponding to the **b** source voltage $V_s$ and the antiferroelectric voltage $V_{AFE}$, **c** current $I$ through the circuit, **d** charge $Q$ on the application of an a.c. voltage pulse $V_s$: $-5 \to +5 \to -5$ V. $R = 3$ kΩ. NC transients are observed at the shadow areas of the regions I and II, which are magnified in **e** and **f**, respectively. The source voltage $V_s$ is shown as the blue line and $V_{AFE}$ transient as the red one in **b**.

d, respectively. In other words, $V_{AFE}$ and $Q$ have opposite variation tendency in the shadow area of region I, suggesting that the antiferroelectric PZO film passes through a NC state. The unstable state during the transition from an antiferroelectric to ferroelectric phase results in a negative differential capacitance in antiferroelectric PZO films. A similar NC effect is observed after $+5 \to -5$ V transition in the shadow area of region II (magnified in Fig. 2f), where the PZO films also undergo the transition from the antiferroelectric to ferroelectric phase. The opposite voltage and charge signals in the regions I and II indicate the NC effect in antiferroelectric PZO films, as highlighted by the shadow. Differently, the voltage and charge transients across the Pt/a-PZO/LSMO capacitor always keep the same variation tendency under the same setup (Supplementary Fig. S3). This phenomenon reveals that the NC effect observed here is intrinsic and not caused by the experimental setup or unexpected sources.

**Distinctive transient NC effect in antiferroelectric films**. A ferroelectric material passes through two regions where the differential capacitance is negative, while switching from one stable polarization to the other and vice versa. But the antiferroelectric materials go through two transitions from antiferroelectric to ferroelectric state, and two transitions from ferroelectric to antiferroelectric state under one cycle of voltage scanning. Given that different switching characteristics of antiferroelectric materials distinguished from ferroelectrics, a different a.c. voltage pulse

sequence of $V_s$: $-5 \to +5 \to +1 \to -5 \to -1 \to +5$ V was applied as input on the same circuit setup to embody all the transition processes in antiferroelectric materials. Here, the voltage of 5 V was applied to switch the antiferroelectric to ferroelectric state ($V_{AFE-FE}$), and the voltage of 1 V was applied to embody the switching process from ferroelectric to antiferroelectric state ($V_{FE-AFE}$) for the antiferroelectric PZO films.

Figures 3a–c displays the transients corresponding to $V_s$, $V_{AFE}$, $I$, and $Q$. It is obvious that the voltage and charge have opposite variation tendency in the shadow areas of regions I–IV, which are magnified in Figs. 3d–g, respectively. Signals in regions I and III occur during the transition from antiferroelectric state to ferroelectric state for antiferroelectric PZO films. Signals in regions II and IV occur after the $+5 \to +1$ V and $-5 \to -1$ V transitions of $V_s$, when switching from two different polarization states (up and down) under high electric filed to the antiferroelectric state, in which net polarization is near zero under a low electric field. Similar effect never appears in the Pt/a-PZO/LSMO capacitor during the whole measurement with the identical procedure and setup (Supplementary Fig. S4), supporting the phenomenon is intrinsic. The NC transients suggest that this effect occurs just during the switching, seriously limiting its isolated use in the low voltage field effect transistors[26]. However, four NC effect in a period of voltage pulse make antiferroelectric materials distinct from traditional NC effect in ferroelectric materials, which extend the time range and increase the number of times. It will further improve the efficiency of voltage

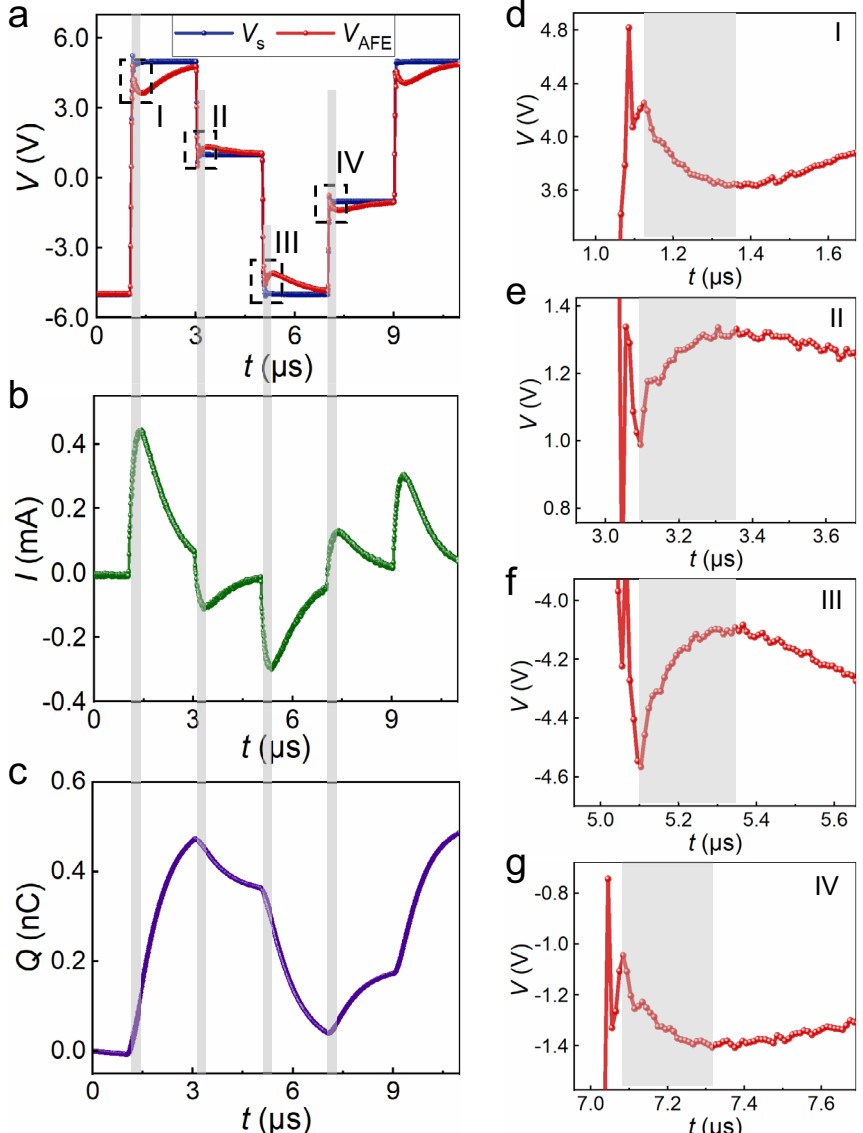

**Fig. 3 Transient responses of an antiferroelectric capacitor under different pulse operation processes.** Transients corresponding to **a** source voltage $V_s$ and the antiferroelectric voltage $V_{AFE}$, **b** current $I$ through the circuit, **c** charge $Q$ on the application of an a.c. voltage pulse $V_s$: $-5 \rightarrow +5 \rightarrow +1 \rightarrow -5 \rightarrow -1 \rightarrow +5$ V. $R = 3\,k\Omega$. NC transients are observed at the shadow areas of regions I, II, III, and IV, which are magnified in **d**, **e**, **f**, and **g** respectively. The source voltage $V_s$ is shown as the blue line and $V_{AFE}$ transient as the red.

amplification or the reductive of power dissipation undoubtedly, if it is put into practical application in the future.

**Negative slope of _P–V_ loop**. The polarization, $P = Q/A - \varepsilon_0 E$, is plotted as a function of $V_{AFE}$ in Fig. 4, which is calculated from the voltage and current transients in Fig. 3. Distinguished from the static NC effect, the transient NC effect occur before the system reaches the switching start point due to the reverse domain nucleation and the accelerated growth, resulting in the irreversible hysteresis loop of polarization $P$ and electric filed $E$[9]. Here, the shape of the loop is decided by the antiferroelectric capacitor because the parasitic capacitance in the circuit cannot induce such a variation. We can observe that the slope of the loop in the regions I–V is negative, indicating the capacitance is negative in these regions. Meanwhile, the hysteresis loop indicates

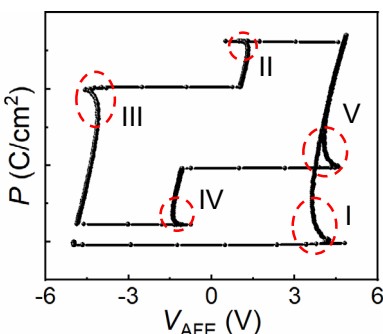

**Fig. 4 Relation between polarization and voltage.** The numbers correspond the transition processes.

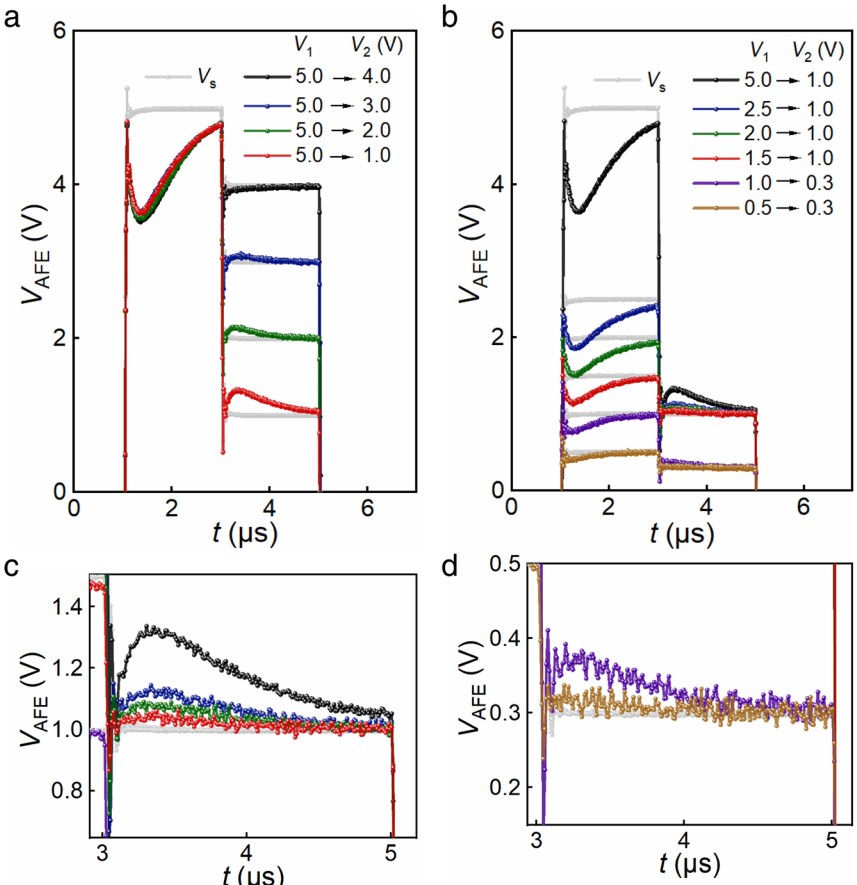

**Fig. 5 $V_{AFE}$ transients under different magnitudes of applied voltages. a** Transient response of source voltage $V_s$ with the same $V_1$ of 5.0 V and different $V_2$, and the corresponding voltage across the antiferroelectric capacitor $V_{AFE}$. **b** Transient response of source voltage $V_s$ with different $V_1$ and $V_2$ of 1.0 or 0.3 V, and the corresponding voltage $V_{AFE}$. The second NC effect in **b** is magnified in **c** and **d**. The gray lines represent the source voltage $V_s$ with two different amplitudes of $V_1$ and $V_2$, and the colored lines represent the transient voltage response under $V_s$.

that the NC regions are not stabilized in the isolated capacitor configuration, and thus the NC effect in this device structure is transient in nature. It is worth noting that the loop is not completely closed owing to the leakage currents, and the polarization is not zero around 0 V because the voltage and charge message near 0 V has not been recorded.

**Relation between NC effect and antiferroelectric switching**. To verify the relationship between the transient NC effect and antiferroelectric switching, different voltage pulses were applied on the series setup by comparing the critical voltage of NC effect, with the switching voltage of antiferroelectric phase transition. Corresponding data on the $V_{AFE}$ transients under different applied voltages is presented in Fig. 5. The first pulse voltage is defined as $V_1$, and the second $V_2$. Keeping $V_1$ as 5.0 V and changing the voltage $V_2$, the first NC effect is not influenced, as shown in Fig. 5a (the original data can be seen in Supplementary Fig. S5). In contrary, $V_2$ strongly affects the second NC effect. The NC effect cannot be seen under $V_2$ of 4.0 V, and begin to appear when the voltage decreases to 3.0 and 2.0 V, and becomes obvious under 1.0 V, which is consistent with the critical voltage of antiferroelectric phase transitions. When $V_1$ is kept as the same, the process of transition from antiferroelectric state to ferroelectric state is not influenced. Just like we have observed in Fig. 5a, the voltage–time ($V_{AFE}$–$t$) curves keep almost unchanged, indicating that the first NC effect under different conditions is not affected, while $V_2$ can greatly affect the process of transition from ferroelectric to antiferroelectric phase. As seen in Fig. 1c, the

transition cannot take place when the voltage is 4.0 V, making NC effect absent. The transition is partial when the voltage is 3.0 and 2.0 V, so that the NC effect is weak. The transition is complete when the voltage is 1.0 V, which is close to the coercive voltage, so that the NC effect become obvious.

As shown in Fig. 5b, c (the original data can be seen in Supplementary Fig. S6), when the $V_1$ of 5.0 V is applied, obvious NC effect occurs and the following $V_2$ of 1.0 V also makes the second NC effect obvious. When $V_1$ decreases, both the first and the second NC effect are getting weaker. As displayed in Fig. 5d, when $V_1$ further decreases to 1.0 V and $V_2$ is kept as 0.3 V, the NC effect are very small, and they disappear when $V_1$ is 0.5 V. The phenomenon is consistent with the phase transition of antiferroelectric films. From Fig. 1c, the switching voltage $V_2$ of 1.0 or 0.3 V is low enough to fully switch the ferroelectric phase to antiferroelectric phase and $V_1$ directly influences the proportion of ferroelectric phase, so both two NC effect are influenced by the voltage $V_1$. The transition from antiferroelectric to ferroelectric phase for the PZO films cannot occur <1.0 V, and thus NC effect are inhibited. When the voltage is <2.5 and >1.0 V, the phase transition is incomplete so that the NC effect have different amplitudes. Obvious NC effect can be seen when $V_1$ are 2.5 and 5.0 V, owing to the fully phase transition from antiferroelectric to ferroelectric phase.

**Physical origin of NC effects in antiferroelectric**. In order to explore the origin of the NC effect microscopically, the scanning transmission electron microscope (STEM) is used to record the

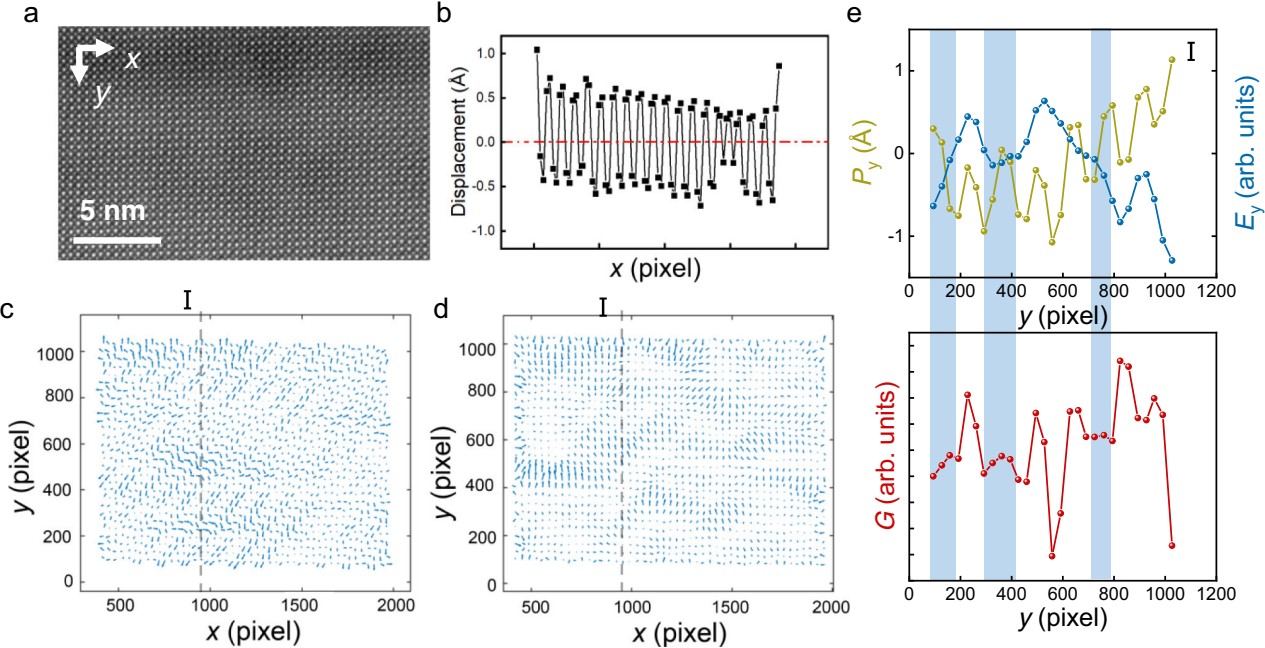

**Fig. 6 Identifying the local regions of NC effect. a** Cross-sectional STEM image of the PZO film. **b** Variation of Pb-cation polar displacement averaged the magnitudes along the [110] direction. **c** Polarization vector **P** mapping extracted from the Pb-cation displacement in **a**. **d** Electric field vector **E** mapping calculated from the surface density of polarization charge $\rho$ in the same region. **e** Variation in the $y$ component of the polarization displacement $P_y$ (yellow line) and electric field $E_y$ (blue line), and the local energy estimated according to $G \approx \int E_y dD_y$ along the perpendicular line I. The light blue shadows in **e** indicate these regions have the negative differential capacitance where $(\partial^2 G)/(\partial^2 D) < 0$ and $dP/dE < 0$.

atom-column information, as displayed in Fig. 6a. The local polarization of the antiferroelectric film PZO can be evaluated by the deviation of the Pb-cation from the geometric center of the four surrounding Zr-site columns. To better compare the Pb displacements, we extract the magnitudes of Pb displacements by averaging the magnitudes of Pb cations along the [110] direction, and plot the spatial variation of the displacements along the $x$-direction in Fig. 6b. The polarization is almost compensated, demonstrating the nearly zero polarization under zero electric field and the antiferroelectricity of the PZO film in a relatively large scale. All of the Pb-cation displacements are presented in the polarization vector maps in Fig. 6c, where two dipole arrangement modes are observed (details in Supplementary Note 1 and Supplementary Fig. S7). The unique polarization configuration should be induced by the film strain and resultant tilting and distortion of the oxygen octahedra[17,27]. Remarkably, the polarization vectors have a fluctuation period of four with a couple of two-layer dipole arrangement, with different magnitudes or directions. And the polarization in the sublattice has the same direction and changes the directions between the sublattices. Without the applied external electric field, the electric filed in the antiferroelectric film arises from the depolarized field. The electric field **E** in the antiferroelectric PZO film is estimated by the surface density of the polarization charge $\rho$ (Supplementary Fig. S8), which is the negative divergence of the polarization **P** (see "Methods" section "Estimation of the depolarized field **E** and free energy $G$"). The electric field vector **E** maps are displayed in Fig. 6d, which have different distribution rules with the polarization **P**.

To build the connection between the polarization and electric field, we plot the $y$ components of the local polarization displacement $P_y$ (yellow line) and the electric field $E_y$ (blue line) along the perpendicular line I (see Figs. 6c, d) in the upper panel of Fig. 6e. And the free energy $G$ evaluated by $G \approx \int E_y dD_y$ ($D_y$ represents the $y$ component of the displacement field **D**, see "Methods" section "Estimation of the depolarized field **E** and free

energy $G$") is presented in the lower panel of Fig. 6e. The differential capacitance $C$ is proportional to $dP/dE$, so the regions where the polarization has the opposite direction to the electric field exhibit NC. Meanwhile, the capacitance is related to $(\partial^2 G)/(\partial^2 D)$, so the regions where the free energy curve has a negative curvature show negative differential capacitance. It is obvious that in Fig. 6e the light blue shadow regions mark the NC regions. Note that several positions in the perpendicular line I possess negative differential capacitance, and this phenomenon is universal in the whole regions (see Supplementary Fig. S9 for more measurements in different regions). The inhomogeneous polarization distribution in PZO film induces the difference of the polarization magnitudes in adjacent sublattices, further leading to the inhomogeneous distribution of the electric field. The polarization and electric field create the regions, where the NC effect emerges. The antiferroelectric materials have denser NC local regions than the ferroelectrics, and the NC effect emerges within the domain and not limited by the domain walls. The total capacitance can be regarded as the series and parallel structures of all the NC regions in three-dimensional space.

## Discussions

In conclusion, the static and transient NC effect are demonstrated from the proof-of-concept experiment in antiferroelectric PZO films. The capacitance enhancement due to static NC effect was observed in the a-HfO$_2$/PZO bilayer capacitor at room temperature. Meanwhile, under pulse measurements, the voltage across the Pt/PZO/LSMO capacitor is found to be decreasing (increasing) with time, while the charge is increasing (decreasing) when the film goes through the phase transitions in a R–C circuit, which provides a direct evidence of the transient NC. Different from the ferroelectric counterpart, four times of NC effect were observed in the antiferroelectric capacitor during a period of voltage pulses, making antiferroelectric materials competitive in the future application. The polarization vector maps and the analysis of depolarization field provide the probable position of

the NC local regions. All these above demonstrate the existence of NC effect in antiferroelectric films, extending the potential applications of NC effect and further reducing the energy dissipation.

## Methods

**Fabrication of antiferroelectric capacitors.** A 50 nm PZO thin film was epitaxially grown on a 20 nm LSMO-buffered STO substrate by pulsed laser deposition (PLD) technique. LSMO film was grown directly on the STO substrate at 670 °C under oxygen partial pressure of 100 mtorr as bottom electrode, and then PZO film was grown at 640 °C, under oxygen partial pressure of 20 mtorr. Afterward, the heterostructure was slowly cooled down at 300 torr of oxygen partial pressure to the room temperature. Pt top electrodes were ex situ deposited by magnetron sputtering at room temperature and patterned using lithographic techniques into square shape with a surface area, $A = 30 \times 30\ \mu m^2$. Paraelectric a-HfO$_2$ film (20 nm) was deposited by magnetron sputtering at room temperature as regular dielectric layer, and a-PZO film was deposited using PLD technique at room temperature as control group.

**Measurements of electric properties.** The $C$–$V$ response was measures by Agilent B1500 at 100 KHz. The voltage pulse was supplied by a pulse generator Agilent B1530, and the current was directly measured by Agilent B1500. During all the measurements, the top Pt electrodes were biased and bottom electrodes LSMO grounded.

**Estimation of the depolarized field E and free energy G.** The surface density of the polarization charge $\rho$ (Supplementary Fig. S8a), the potential $\psi$ (Supplementary Fig. S8b), the depolarized field $\mathbf{E}$ (Fig. 6d), the displacement field $\mathbf{D}$ (Supplementary Fig. S8c), and the free energy $G$ are estimated by the following formulas, respectively:

$$\rho = -\mathrm{div}(\mathbf{P}) \tag{1}$$

$$\psi = \frac{1}{4\Pi\varepsilon_0} \int \frac{\rho(r')}{|r-r'|} \mathrm{d}S \tag{2}$$

$$\mathbf{E} = \nabla\psi \tag{3}$$

$$\mathbf{D} = \varepsilon_0 \mathbf{E} + \mathbf{P} \tag{4}$$

$$G = \int E_y \mathrm{d}D_y \tag{5}$$

## Data availability
The data that support the findings of this study are available from the corresponding author upon reasonable request.

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

## Acknowledgements
We acknowledge the help by Dr. Jian Liu, YueZe Tan, Prof. Jiamian Hu, and Minyi Dai. This work has been supported by the National Key R&D Program of China (Grant Nos. 2016YFA0203800 and 2017YFB0405604), the National Natural Science Foundation of China (Grant No. 51871130), and the Natural Science Foundation of Beijing, China (Grant No. JQ20010).

## Author contributions
C.S. and F.P. conceived and supervised the project. L.L.Q. and M.U.F. fabricated the devices. L.L.Q., T.Q.L., and T.-L.R. performed the electrical measurements. C.S., L.L.Q., S. Q.Y., and Y.M.S. performed the data analysis and co-wrote the manuscript. L.L.Q., C.C., and H.P.X. performed the calculation of the electric field and free energy. All the authors discussed the results and revised the manuscript.

## Competing interests
The authors declare no competing interests.

## Additional information
**Suppl ementary information** The online version contains supplementary material available at https://doi.org/10.1038/s41467-021-24530-w.

