## [Peer Review File · Nature Communications]

REVIEWER COMMENTS

Reviewer #1 (Remarks to the Author):

This manuscript reports on negative capacitance in antiferroelectric PbZrO₃ film. Experiments are reasonable and corresponding results are expected. However, there are still some problems with the manuscript as follows:

1. "Static negative capacitance (NC)" and "transient NC" are mentioned many times in the article, is this statement accurate? Are there any convincing claims on "static NC" that have been reported?
2. The microscopic physical mechanism of negative capacitance is that "when the ferroelectric transforms from one polarization to the other around $Q = 0$, it goes through a region, where the reciprocal of second derivation of U and Q , that is, $C = [d^2U/dQ^2]^{-1}$ is negative". If this is regarded as the "transient NC effect", what is the microscopic physical mechanism of the "static NC" effect?
3. Figure 2 characterizes the XRD and small signal C-V of PbZrO₃ film. It is recommended to add a hysteresis loop test (polarization-electric field, P-E) to further visual verify the good antiferroelectric properties of the film.
4. In addition, there are some minor problems in language and format that need to be checked and modified.

Reviewer #2 (Remarks to the Author):

This paper shows both static and transient Negative Capacitance effect in anti-ferroelectric PZO. This is a new material system for exploring negative capacitance effect. Most of the experimental results provided in the manuscript are also satisfactory. However, before acceptance for publication, I suggest the authors clarify the following points:

- 1) For the static NC results with a-HfO₂/PZO/LSMO experiments, the authors need to provide frequency dependent results, especially properly investigating Maxwell-Wagner effects, quantifying how much these effects are contributing to the observed capacitance enhancement.
- 2) The authors also need to provide a clear description of all the thickness used in their stacks in the main text. In addition, they should mention what is the effective permittivity that they are observing in their film such as PZO/LSMO and a-HfO₂ and compare them to what is known from literature.

Reviewer #3 (Remarks to the Author):

Authors study the effect of Negative Capacitance (NC) in the capacitor containing the antiferroelectric compound, PbZrO₃ as a working material. The article considers two types of the NC, the time-dependent transient one, and the thermodynamically stable, static NC. As concerns the transient NC, this effect was extensively measured and explained in various materials, hence the author's advancement for the case of PbZrO₃ has only a slightly incremental character. As concerns the static one, the by far more interesting proof of concept could present the real challenge, however, the authors' arguments are based only on the experimental measurements, grounded on the interpretation of the widely-circulated but a priori self-contradictory concept that the static NC can occur in the monodomain capacitor with electrodes.

More specifically, the authors state that "Capacitance enhancement has been widely adopted to observe the static NC effects in ferroelectric heterostructure and superlattice [5,6,16,17]". However, references [6,16,17] from this list are irrelevant since they deal with NC in the Para-Ferro

superlattices, where the Ferro-layer is not covered by electrodes, the case different from that, described in the article. The reference [5], as well as an explication of Fig.1, are oversimplified and incorrect in its interpretation, since, as was previously shown in [Landauer, R. Can capacitance be negative? Collect. Phenom. 2, 167–170 (1976)] the considered in [5] monodomain structure with electrodes is always unstable towards splitting onto domains, that completely change the electric properties of the capacitor (NB: Article of Landauer is rarely available, for detailed explication one can consult also Communication Physics, 2, 22, (2019)).

I believe that publication of the article is possible only in the case when authors quantitatively justify their observation, basing on the correct, non-self-contradictory description of the static NC.

Igor Lukyanchuk

We appreciate three Referees' careful review and positive evaluations, for example, Reviewer #1: "Experiments are reasonable and corresponding results are expected." Reviewer #2: "This is a new material system for exploring negative capacitance effect. Most of the experimental results provided in the manuscript are also satisfactory." and Reviewer #3: "I believe that publication of the article is possible only in the case when authors quantitatively justify their observation, basing on the correct, non-self-contradictory description of the static NC." And their constructive comments help us to improve our manuscript further. The detailed point-by-point responses to the reviewers' comments are summarized below, where amendments of our revised manuscript are in purple color.

The main modifications include:

1. We clarified the concepts "static negative capacitance (NC)" and "transient NC" in the introduction part and added some references to verify the accuracy of these definitions.
2. We explained and distinguished the microscopic physical mechanism behind static and transient NC effects in introduction part and results part.
3. The hysteresis loop for PbZrO_3 (PZO) film was obtained to verify the antiferroelectric performance and added to Fig. 1b.
4. We performed the frequency dependence tests of capacitance to exclude the influence of Maxwell-Wagner effects.
5. The effective permittivity of PZO and a- HfO_2 films was calculated and compared with the reported data.
6. We performed the HAADF-STEM to mapping the polarization vectors in the antiferroelectric PZO film to explain why antiferroelectric film has the nature of negative capacitance effects and justify our observation.
7. Minor problems of language and format were improved.

Response to Reviewer #1:

This manuscript reports on negative capacitance in antiferroelectric PbZrO_3 film. Experiments are reasonable and corresponding results are expected.

Author reply: We are grateful to the reviewer for the positive evaluation of our work.

Q1. “Static negative capacitance (NC)” and “transient NC” are mentioned many times in the article, is this statement accurate? Are there any convincing claims on “static NC” that have been reported?

Author reply: The concept “static NC” was firstly proposed by Salahuddin and Datta in 2008 that the ferroelectric capacitance C_{FE} is a function of the polarization P , where C_{FE} is negative only in a limited region of P and would have a minimum for $P = 0$ (Ref. 3). This theoretically hysteresis-free NC is called as static NC. After that, a lot of works have been done to interpret static NC from proof-of-concept and device-implementation aspects (Refs. 6–8). The demonstration of static NC from the concept is realized by connecting a paraelectric film with a ferroelectric layer. Referring to the device implementation, the ferroelectric film is added into transistors for reducing the subthreshold swing to make use of the NC effects of ferroelectric materials.

The transient NC was initially detected in 2015 that the voltage and charge across the ferroelectric capacitor have opposite variation trends during the ferroelectric switching (Ref. 10). This hysteresis NC is defined as the transient NC effects. The demonstration of transient NC is realized by exposing the internal node of an MFIS structure or connecting a ferroelectric capacitor in series with an external resistor or capacitance, and monitoring the real-time voltage and charge responses across the ferroelectric capacitor.

These two types of NC effects have been clearly distinguished in the chapter 10 of the book *Ferroelectricity in doped Hafnium Oxide* (Ref. 25) and the review article *Ferroelectric Negative Capacitance* (Ref. 5). To clarify the concepts more clearly, we amended the statement in the Introduction section as (Introduction, Page 2) “The NC effects are usually classified into static NC effects and transient NC effects.⁵ The static NC effects are theoretically hysteresis-free³ and can be stabilized by adding a positive capacitance in series.^{6–8} The transient NC is always accompanied by hysteresis. The voltage drop across the ferroelectric is observed in the limited time range by increasing the charge Q when the pulse is applied in a circuit with a capacitor in series with an external resistor (R – C circuit).^{9,10}”.

Q2. The microscopic physical mechanism of negative capacitance is that “when the ferroelectric transforms from one polarization to the other around $Q = 0$, it goes through a region, where the reciprocal of second derivation of U and Q , that is, $C = [d^2U/dQ^2]^{-1}$ is negative”. If this is regarded as the “transient NC effect”, what is the microscopic physical mechanism of the “static NC” effect?

Author reply: Negative capacitance effects originate from the negative second derivation of potential energy U and charge Q in the Landau energy landscape

or the negative branch of the curve of Q and voltage V with double-S shape. This is the thermodynamic origin of both static and transient NC effects. The difference of switching dynamics distinguishes the static NC and transient NC.

The NC regions are unstable for the local maximum energy so that static NC is unreachable in voltage driven devices. Whereas setting charge Q as driving force can reverse the $Q-V$ curve from multi-valued double-S shape to single-valued double-N shape, which makes static NC stabilized in the negative region (Ref. 5). And the direct way to control the charge is to connect a ferroelectric or antiferroelectric capacitor with a paraelectric capacitor (Ref. 3).

Under the electric bias, the switching instability may occur before the system reaches the phase switching start point in the S-shape $Q-V$ curves due to the reverse domain nucleation and the accelerated growth (Ref. 26). The instability turns the non-hysteresis $Q-V$ curve with double-S shape into irreversible hysteresis loop. The negative slope at the beginning of the switching corresponds to the transient NC. The difference between static and transient NC effects have been emphasized in the manuscript (Results, Negative slope of $P-V$ loop, Page 12) as “Distinguished from the static NC effects, the transient NC effects occur before the system reaches the switching start point due to the reverse domain nucleation and the accelerated growth, resulting in the irreversible hysteresis loop of polarization P and electric field E .²⁶”

Q3. Figure 2 characterizes the XRD and small signal $C-V$ of PbZrO_3 film. It is recommended to add a hysteresis loop test (polarization-electric field, $P-E$) to further visual verify the good antiferroelectric properties of the film.

Author reply: According to the reviewer’s suggestion, the hysteresis loop of antiferroelectric PZO film was measured to demonstrate the antiferroelectricity of the film used in our work. The hysteresis loop for PZO film shows large saturation polarization strength of $50 \mu\text{C cm}^{-2}$. It is worth noting that the switching electric field is consistent with the results of $C-V$ tests. The little mismatch comes out from the difference of the two measurements. The figure is added into **Fig. 2b Antiferroelectric polarization P as a function of electric field E** . The corresponding description is added in the manuscript (Results, Capacitance enhancement induced by NC effects in the antiferroelectric capacitor. Page 3 and 5) as “In the meanwhile, large saturation polarization strength of $50 \mu\text{C cm}^{-2}$ and weak remanent polarization of PZO film in Fig. 2b further demonstrate the antiferroelectricity. The weak remanent polarization most likely originates from the compressive strain induced by SrTiO_3 substrate,

producing the local ferroelectricity.¹⁶” and “The position of each peak is consistent with the coercive electric field of the P – E loop. The antiferroelectric phase is transformed to ferroelectric phase when the applied electric field is over $\sim 0.45 \text{ MV cm}^{-1}$, and the ferroelectrics are recovered to antiferroelectrics when the electric field is lower than $\sim 0.2 \text{ MV cm}^{-1}$.”.

Fig. 2b Antiferroelectric polarization P as a function of electric field E .

Q4. In addition, there are some minor problems in language and format that need to be checked and modified.

Author reply: We have carefully checked and modified the language and format problems.

Response to Reviewer #2:

This paper shows both static and transient Negative Capacitance effect in anti-ferroelectric PZO. This is a new material system for exploring negative capacitance effect. Most of the experimental results provided in the manuscript are also satisfactory.

Author reply: We appreciate Reviewer’s support and the positive evaluation of our work.

However, before acceptance for publication, I suggest the authors clarify the following points:

Q1. For the static NC results with a-HfO₂/PZO/LSMO experiments, the authors need to provide frequency dependent results, especially properly investigating

Maxwell-Wagner effects, quantifying how much these effects are contributing to the observed capacitance enhancement.

Author reply: As advised by the reviewer, the frequency dependence of capacitance of another a-HfO₂/PZO sample is measured to rule out the contribution of Maxwell-Wagner effect on the capacitance enhancement, and the results are displayed in Supplementary Fig. 1 *Capacitance as a function of frequency in Pt/a-HfO₂/LSMO and Pt/a-HfO₂/PZO/LSMO capacitors*. The Maxwell-Wagner effect can cause the dielectric permittivity enhancement in superlattice structures, and it is proved to be related with the number of heterojunction interfaces and measurement frequency (Ref. 20). The single interface in our work contributes little to the Maxwell-Wagner effect. Moreover, the measurement frequency of 100 KHz is usually used to judge the capacitance enhancement effects in the literatures previously reported (Ref. 6). We have added the discussion about this effect in the manuscript (Results, Capacitance enhancement induced by NC effects in the antiferroelectric capacitor. Page 6) as “The Maxwell-Wagner effect also can lead to the enhancement of effective permittivity due to the accumulation of charges at the interface of heterostructures or superlattices when the electric field is applied. This effect is also related closely with the number of heterojunction interfaces and the measurement frequency.²⁰ Note that this effect has little influence on the capacitance enhancement due to the single interface in our structures. And the capacitance of a-HfO₂/PZO sample is indeed higher than that of corresponding a-HfO₂ sample through the whole frequency range (Supplementary Fig. S1). The capacitance enhancement effect demonstrates that the NC effects originate from the negative capacitance effect of the antiferroelectric film, rather than the Maxwell-Wagner effect.”.

Supplementary Fig. 1. Comparison of the capacitance as a function of frequency in Pt/a-HfO₂/LSMO and Pt/a-HfO₂/PZO/LSMO capacitors. To

exclude the Maxwell-Wagner effect and to show the reproducibility, the frequency dependent capacitance of another a-HfO₂/PZO sample was measured. Corresponding value is higher than its a-HfO₂ counterpart through the whole frequency range when the frequency is swept from 1 KHz to 1 MHz.

Q2. The authors also need to provide a clear description of all the thickness used in their stacks in the main text. In addition, they should mention what is the effective permittivity that they are observing in their film such as PZO/LSMO and a-HfO₂ and compare them to what is known from literature.

Author reply: We have emphasized all the thickness used in the stacks in the manuscript (Results, Capacitance enhancement induced by NC effects in the antiferroelectric capacitor. Pages 5 and 6) as “To demonstrate the static NC effects in antiferroelectric PZO films, two capacitors Pt(10 nm)/amorphous HfO₂(a-HfO₂, 20 nm)/La_{2/3}Sr_{1/3}MnO₃(LSMO, 20 nm), Pt(10 nm)/a-HfO₂(20 nm)/PZO(50 nm)/LSMO(20 nm) were designed to compare the values of their capacitance. For a comparison, Pt(10 nm)/a-HfO₂(20 nm)/LSMO(20 nm) and Pt(10 nm)/a-HfO₂(20 nm)/a-PZO(50 nm)/LSMO(20 nm) capacitors were fabricated to verify the validity of the capacitance enhancement.”

The dielectric relative permittivity of the PZO film is calculated and compared in the manuscript (Results, Capacitance enhancement induced by NC effects in the antiferroelectric capacitor. Page 5) as “The effective permittivity of 50 nm PZO film calculated from the C–V loop is 61.33 in our work, which is lower than that of bulk PZO with 220 at the frequency of 100 KHz.¹⁹ The relative permittivity of 20 nm amorphous HfO₂ film is calculated as 13.67 at the same frequency, which is comparable with the reported value. The permittivity is closely related to the crystal structure, growth quality, film thickness, measurement frequency, temperature and so on.¹⁹ Thus the difference of dielectric permittivity most likely results from the material system and the film quality.”

Response to Reviewer #3:

Authors study the effect of Negative Capacitance (NC) in the capacitor containing the antiferroelectric compound, PbZrO₃ as a working material. The article considers two types of the NC, the time-dependent transient one, and the thermodynamically stable, static NC. As concerns the transient NC, this effect was extensively measured and explained in various materials, hence the author's advancement for the case of

PbZrO₃ has only a slightly incremental character. As concerns the static one, the by far more interesting proof of concept could present the real challenge, however, the authors' arguments are based only on the experimental measurements, grounded on the interpretation of the widely-circulated but a priori self-contradictory concept that the static NC can occur in the monodomain capacitor with electrodes.

More specifically, the authors state that "Capacitance enhancement has been widely adopted to observe the static NC effects in ferroelectric heterostructure and superlattice [5,6,16,17]". However, references [6,16,17] from this list are irrelevant since they deal with NC in the Para-Ferro superlattices, where the Ferro-layer is not covered by electrodes, the case different from that, described in the article. The reference [5], as well as an explication of Fig.1, are oversimplified and incorrect in its interpretation, since, as was previously shown in [Landauer, R. Can capacitance be negative? Collect. Phenom. 2, 167–170 (1976)] the considered in [5] monodomain structure with electrodes is always unstable towards splitting onto domains, that completely change the electric properties of the capacitor (NB: Article of Landauer is rarely available, for detailed explication one can consult also Communication Physics, 2, 22, (2019)).

I believe that publication of the article is possible only in the case when authors quantitatively justify their observation, basing on the correct, non-self-contradictory description of the static NC.

Author reply: Thanks for the careful review and helpful suggestions. To clarify the device structure more clearly, the corresponding sentences have been modified in the manuscript (Results, Capacitance enhancement induced by NC effects in the antiferroelectric capacitor. Pages 5 and 6) as "To demonstrate the static NC effects in antiferroelectric PZO films, two capacitors Pt(10 nm)/amorphous HfO₂(a-HfO₂, 20 nm)/La_{2/3}Sr_{1/3}MnO₃(LSMO, 20 nm), Pt(10 nm)/a-HfO₂(20 nm)/PZO(50 nm)/LSMO(20 nm) were designed to compare the values of their capacitance. For a comparison, Pt(10 nm)/a-HfO₂(20 nm)/LSMO(20 nm) and Pt(10 nm)/a-HfO₂(20 nm)/a-PZO(50 nm)/LSMO(20 nm) capacitors were fabricated to verify the validity of capacitance enhancement.". The antiferroelectric film is in series with the dielectric layer and is not directly covered by the metallic electrode in our work. The multidomain structure induced by depolarization field is responsible for the observation of negative capacitance effect due to the large screen length in the oxide electrode.

Phase field modeling may be an effective tool to reveal the mechanism of negative capacitance effects. Several papers have been reported to use phase field method to model the negative capacitance effects in ferroelectric

materials (Ref. 26, I. Luk'yanchuk et al. *Commun. Phys.* **2**, 22 (2019)). But the application of phase field modeling in antiferroelectric films remains some problems. The electric field is not coupled with the antiferroelectric order parameter in most models about antiferroelectric materials, which results in the competitive relation of ferroelectric and antiferroelectric order parameters. The change in electric displacement cannot be achieved from the models. On the other hand, the order parameter in models does not reflect the realistic atomic displacement and properties behind the structures. Both the problems limit the analysis of negative capacitance effects via simulation. Phase field modelling will be an effective tool to further deepen our understanding to the negative capacitance effects in antiferroelectric materials if both the problems are resolved. That will attract more researchers to improve the existing models.

Alternately, spatially resolved polar vector and electric field mapping is also an effective method to explore the origin and mechanism of negative capacitance effects. Yadav et al. (Ref. 11, Yadav, A. K. et al. *Nature* **565**, 468–471 (2019)) reported in 2019 that the local regions of negative capacitance effects emerge at the domain walls where the polarization is suppressed and the energy density is higher than that in bulk of the domains. In order to justify the NC effects in antiferroelectric film, the atomic-resolution high resolution scanning transmission electron microscope with a high angle annular dark field (HAADF-STEM) images of PZO cross-section were obtained and analyzed. Different from the ferroelectrics, the polarization suppression occurs at the sublattices in antiferroelectric film. These sublattices could lead to the emergence of negative capacitance. The relevant statement has been modified in the manuscript (Results, Capacitance enhancement induced by NC effects in the antiferroelectric capacitor. Page 7) as “Referring to the physical origin of NC effects in antiferroelectric films, polarization vector mapping of the PZO film based on scanning transmission electron microscope (STEM) is performed and the sublattices where the polarization is suppressed are thought as the probable local NC regions (Supplementary Fig. S6 and note 1).” The corresponding results and discussions are added in Supplementary material note 1, as follows:

In order to further connect negative capacitance effects with internal dipole arrangements of antiferroelectric films, the Pb-cation displacement vector mappings that represent the local polarization, were performed via scanning transmission electron microscope with a high angle annular dark field (HAADF-STEM). The off-center displacements of Pb cations are evaluated by the deviation from the geometric center of four surrounding B-site columns. In our sample, the typical antiparallel dipole arrangements are observed along

[110] direction with commensurate modulations of $\frac{1}{4}\{110\}$, as shown in Fig. S6a and 6b. The different colors in Fig. S6b represent the different polarization directions, as shown in the scale bar. To better compare Pb displacements, we extract the magnitudes of Pb displacements from the vector maps in Fig. S6b by averaging the magnitudes of 27 or 12 layers of Pb cation along the length direction of the stripes. The polarization magnitudes are plotted in Fig. S6c, 6d and 6e, which display the polarization vector distribution through the whole black rectangle, the upper shadow region, and the lower shadow region respectively. In Fig. S6c, the polarization due to Pb displacements fluctuates like wavelike curves across the modulations. This variation fashion of sinusoidal shape has been previously reported by MacLaren *et al.*²⁷ and Tan *et al.*²⁸. The polarization through the whole regions is almost compensated, demonstrating the antiferroelectricity of PZO film. Whereas the polarization in the upper shadow region tends to align upper left, and the lower right vectors have small magnitudes, producing suppressed polarization regions at the sublattices, as shown in Fig. S6d. Differently, the polarization vectors in the lower shadow region tend to align in the direction of lower right, inducing the polarization suppression regions at the sublattice with upper left polarization direction, as shown in Fig. S6e.

Compared with the polarization vector variation in ferroelectric materials, antiferroelectric materials have a similar variation tendency, but with much shorter period. It has been reported that the suppressed polarization in the domain wall leads to negative capacitance in ferroelectrics.¹¹ In antiferroelectric films, the adjacent sublattices have opposite polarization directions with the same magnitudes theoretically. In fact, the local regions always cannot compensate fully though the polarization can be compensated in the whole film or greater regions. One of the adjacent sublattices has greater polarization magnitudes and the other is suppressed. These polarization suppression regions at the sublattices could be the position where negative capacitance effects emerge. More negative capacitance regions exist in antiferroelectric materials, guaranteeing the stronger negative capacitance effects in antiferroelectrics than that their ferroelectric counterparts. Deeper understanding on the negative capacitance in antiferroelectrics needs further studies.

Supplementary Fig. 6. Identifying the regions of NC effects. (a) A cross-sectional HAADF-STEM image of the PZO film. (b) Representative polarization vector mapping of Fig. S6a. The different colors represent the directions of polarization vectors, as shown in the scale bar. (c)–(e) Pb-displacement magnitudes extracted from the Pb-displacement vector mappings in the whole rectangle region (c), the upper shadow region (d), and the lower shadow region (e).

Reviewers' comments:

Reviewer #1 (Remarks to the Author):

I appreciate for the revision. I now suggest that the revised manuscript is suitable for publication in Nature

Reviewer #2 (Remarks to the Author):

The authors have answered my comments satisfactorily. Looking at the overall rebuttal, I am happy with their responses. I recommend publication of this paper.

Reviewer #3 (Remarks to the Author):

Unfortunately, the authors fail to satisfactory answer my concerns concerning the physical origin and explication of the phenomena they observed, which cast doubt on the credibility of the observation of the static negative capacitance.

More specifically, their explanation, that stays without changing, is based on the model of the monodomain sample, depicted in Fig. 1. As was indicated in my previous report this model is self-contradictory because of the instability towards decay on the multidomain state. Although in their rebuttal letter authors mention that "The antiferroelectric film is in series with the dielectric layer and is not directly covered by the metallic electrode in our work. The multidomain structure induced by depolarization field is responsible for the observation of negative capacitance effect due to the large screen length in the oxide electrode", the corresponding explication and validation of the given above statement is not incorporated in the text of the article.

Moreover, the authors claim the difficulties with the explication of the phenomena they observed: "Both the problems limit the analysis of negative capacitance effects via simulation" and "deeper understanding of the negative capacitance in antiferroelectrics needs further studies".

In the absence of the appropriate qualitative and non-self-contradictory physical explanation of the origin of the observed phenomena and, as a consequence, of at least an approximate evaluation and comparison of the parameters of the negative capacitance for their experimental setup, I can not recommend this article for publication.

We appreciate Reviewer 1# “I appreciate for the revision. I now suggest that the revised manuscript is suitable for publication in Nature Communications”, and Reviewer 2# “The authors have answered my comments satisfactorily. Looking at the overall rebuttal, I am happy with their responses. I recommend publication of this paper”. Although Reviewer 3# gave a negative conclusion, we are grateful to his/her time and constructive comments on our work. We think he/she sets a very high requirement for the first work on the negative capacitance in antiferroelectrics. We also address his/her comments carefully with several months’ work.

The main modification includes:

1. We have clarified that the monodomain model in Fig. 1 is just to introduce the concept of the negative capacitance in antiferroelectrics for simplicity.
2. We have studied the physical origin of the negative capacitance effect specifically in the antiferroelectric film based on the polarization vector mapping and concomitant electric field analysis.

Response to Reviewer #1

Remarks: I appreciate for the revision. I now suggest that the revised manuscript is suitable for publication in Nature Communications.

Author Reply: Thanks a lot for the positive evaluation.

Response to Reviewer #2

Remarks: The authors have answered my comments satisfactorily. Looking at the overall rebuttal, I am happy with their responses. I recommend publication of this paper.

Author Reply: We are grateful to the positive evaluation.

Response to Reviewer #3:

Comments: Unfortunately, the authors fail to satisfactory answer my concerns concerning the physical origin and explication of the phenomena they observed, which cast doubt on the credibility of the observation of the static negative capacitance.

More specifically, their explanation, that stays without changing, is based on the

model of the monodomain sample, depicted in Fig. 1. As was indicated in my previous report this model is self-contradictory because of the instability towards decay on the multidomain state. Although in their rebuttal letter authors mention that “The antiferroelectric film is in series with the dielectric layer and is not directly covered by the metallic electrode in our work. The multidomain structure induced by depolarization field is responsible for the observation of negative capacitance effect due to the large screen length in the oxide electrode”, the corresponding explication and validation of the given above statement is not incorporated in the text of the article.

Moreover, the authors claim the difficulties with the explication of the phenomena they observed: “Both the problems limit the analysis of negative capacitance effect via simulation” and “deeper understanding of the negative capacitance in antiferroelectrics needs further studies”.

In the absence of the appropriate qualitative and non-self-contradictory physical explanation of the origin of the observed phenomena and, as a consequence, of at least an approximate evaluation and comparison of the parameters of the negative capacitance for their experimental setup, I cannot recommend this article for publication.

Author Reply: We appreciate the reviewer’s constructive comments. The energy landscape in Fig. 1 is used to show the concept of the negative capacitance (NC) effect in antiferroelectrics for simplicity, which is not conflict with the multidomain structures observed in our samples. The polarization vector mapping recorded by STEM and electric field analysis reveals that the opposite variation trends of the polarization vector and electric field between the adjacent sublattices produce the negative dP/dE , making the capacitance negative.

In our samples, the antiferroelectric PbZrO_3 (PZO) film is sandwiched by the amorphous HfO_2 film and the oxide electrode SrRuO_3 , which favors the formation of the multidomain structures. And indeed, the multidomain structures of our samples have been observed in the polarization vector mapping recorded by the scanning transmission electron microscope, as shown in Fig. S1. The monodomain model in Figure 1 is used to exhibit a concept of the negative capacitance (NC) effect in antiferroelectrics from the Energy landscape viewpoint, for simplicity. This is a common way for showing the NC effect in ferroelectrics with multidomain [such as Ref. 19, Nature 534, 524–518 (2016)]. The explanation and demonstration of NC effect are not based on the monodomain theory. We have added corresponding statements in Page 3: Although this monodomain model can serve as the basis for the

interpretation of the NC effect in antiferroelectric film for simplicity,¹⁵ it cannot describe the true structures of the antiferroelectric film. The multidomain structure is favored due to the competition of the domain wall energy and the elastic energy in antiferroelectrics.¹⁶ Also, the multidomain structures have been observed in our samples (Supplementary Fig. S1). The NC effect discussed below is based on the multidomain structures.

In order to understand the observed phenomenon, the polarization vector mapping and electric field analysis are performed to trace the origin of the NC effect in antiferroelectrics. The antiferroelectric film has rich sublattice structures where the polarization is reversed. Meanwhile, the depolarization field is opposite to the polarization direction without external field in these regions, making dP/dE (proportional to the capacitance C) negative. All the regions where the polarization directions along the out-of-plane are reversed can be the local NC regions. Corresponding experimental data (Fig. 7) and discussions are added to Pages 17 and 18: In order to explore the origin of the NC effect microscopically, the scanning transmission electron microscope (STEM) is used to record the atom-column information, as displayed in Fig. 7a. The local polarization of the antiferroelectric film PZO can be evaluated by the deviation of the Pb-cation from the geometric center of the four surrounding Zr-site columns. All the Pb-cation displacements are then presented in the polarization vector maps in Fig. 7b, where two dipole arrangement modes are observed (Supplementary Fig. S7). To better compare the Pb displacements, we extract the magnitudes of Pb displacements from the vector maps by averaging the magnitudes of Pb cations along the [110] direction and plot the spatial variation of the displacements along the x -direction in Fig. 7c. Remarkably, the polarization vectors have a fluctuation period of 4 with a couple of two-layer dipole arrangement with different magnitudes or directions. The polarization due to Pb displacements is almost compensated, demonstrating the antiferroelectricity of the PZO film in a relatively large scale. To build the connection between polarization and the electric field, we plot the y components of the local polarization (also refers to as the direction of the electric field) along the x -direction in the upper and lower areas of Fig. 7b in Figs. 7d and 7e respectively. It is obvious that the y components of the polarization for the adjacent sublattices have different directions in the most regions. And the polarization vectors change the directions between the sublattices.

Without the applied external electric field, the electric field in the antiferroelectric film arises from the depolarized field. Thus, the internal electric field tends to have the opposite direction to the polarization direction, which induces the opposite variation trends of the electric field to the polarization

direction. The capacitance C is proportional to dP/dE , which decides that the NC local regions can be between the adjacent sublattices where the polarization and electric field happen to change from negative to positive or from positive to negative. On the other hand, according to the formula $D = \epsilon_0 E + P$, $U = \int E dD$ and $C \propto \frac{\partial^2 U}{\partial^2 D}$ (D represents the displacement field), the zero electric field position would be the extreme large point of the energy density, which means that the second derivation of the free energy G and the displacement field D is negative around the zero electric field. All the regions between the adjacent sublattices can produce the NC effect. The total capacitance can be regarded as the series and parallel structures of all the NC regions in three-dimensional space.

Fig. 7 Identifying the regions of NC effect. **a** Cross-sectional STEM image of the PZO film. **b** Polarization vector mapping extracted from the Pb-cation displacement of Fig. 7a. **c** Pb-displacement magnitudes extracted from the Pb-displacement vector maps in the whole region along the [110] direction. **d** and **e** show the y components of the Pb-displacement magnitudes in the upper and lower region of **b**, respectively.

REVIEWER COMMENTS

Reviewer #3 (Remarks to the Author):

In the new version of the manuscript authors seriously accounted for my remarks concerning the revealing nature of the static negative capacitance (NC) they observed. In particular, they provided additional measurements and discovered that the AFE order parameter texture is highly nonuniform and the component of polarization emerges because of this inhomogeneity. They ascribed the observed static NC to the nonuniform depolarization fields, arising due to spatial fluctuations of the order parameter.

However, despite the remarkable efforts of authors and very interesting observation of the inhomogeneity of the polarization I still consider the justification of the origin of the NC as unconvincing. More specifically, the relation of the observed spatial fluctuation of the depolarization fields and the NC is explained in vague and unclear terms. However, I consider the new results to be very interesting and encourage authors to make more efforts to reveal the physics of the NC phenomena.

This can be done, for example, in either of two ways. (i) Conduct the additional in-field measurements of the polarization distribution at the same region. After subtracting the zero-field and in-field results the local capacitance distribution can be extracted and drawn. Then the sign of the total capacitance, obtained by integration over the sample volume should be examined. (ii) Provide the simple mathematical model that explains the origin of the NC from the observed inhomogeneities.

In addition, I have a few minor remarks.

a) I have a feeling that the observed inhomogeneities are related to the compensation of the internal strain, like ferroelastic domains in strained ferroelectric films. Is this correct?

b) The explication of the static NC by figure 1 is still misleading. In fact, in Fig.1 the system passes through the "forbidden maximum" at a finite field, hence the related NC has the differential finite-voltage meaning, $C=dU/dQ$. What authors claim is to be observed is the real static $C=U/Q<0$ arising at $U=0$. Appeal to Fig. 1 can only hide the genuine origin of this NC (if the effect is valid/correct).

c) It would be instructive to plot the depolarization charge distribution $\rho=-\text{div}(P)$, related to Fig. 7.

Igor Lukyanchuk

We are grateful to the positive evaluations for our new version by the reviewer: “the remarkable efforts of authors and very interesting observation of the inhomogeneity of the polarization.” And the constructive comments indeed help us to understand the origin of the NC effect. The detailed point-by-point responses to the reviewers’ comments are summarized below:

1. We have presented a mathematic model to explain the origin of the NC effect.
2. The tilt and distortion induced by the strain are responsible for the inhomogeneous polarization observed in the PbZrO_3 (PZO) film.
3. Figure 1 is deleted to avoid the misleading about monodomain and multidomain models.
4. The depolarization charge distribution is plotted in Supplementary Fig. 8a.

Response to the Reviewer:

Comments: In the new version of the manuscript authors seriously accounted for my remarks concerning the revealing nature of the static negative capacitance (NC) they observed. In particular, they provided additional measurements and discovered that the AFE order parameter texture is highly nonuniform and the component of polarization emerges because of this inhomogeneity. They ascribed the observed static NC to the nonuniform depolarization fields, arising due to spatial fluctuations of the order parameter.

However, despite the remarkable efforts of authors and very interesting observation of the inhomogeneity of the polarization I still consider the justification of the origin of the NC as unconvincing. More specifically, the relation of the observed spatial fluctuation of the depolarization fields and the NC is explained in vague and unclear terms. However, I consider the new results to be very interesting and encourage authors to make more efforts to reveal the physics of the NC phenomena.

This can be done, for example, in either of two ways. (i) Conduct the additional in-field measurements of the polarization distribution at the same region. After subtracting the zero-field and in-field results the local capacitance distribution can be extracted and drawn. Then the sign of the total capacitance, obtained by integration over the sample volume should be examined. (ii) Provide the simple mathematical model that explains the origin of the NC from the observed inhomogeneities.

Author Reply: We appreciate Prof. Igor Lukyanchuk's constructive comments and kind suggestions. The inhomogeneous polarization is observed in the antiferroelectric PZO film, which should be responsible for the NC effect. To clarify the origin of the NC effect from the observed inhomogeneous polarization, the surface density of the polarization charge ρ , electric potential ψ , depolarization field E and the free energy G are calculated in the same region, as shown in Fig. 6. Accordingly, we add a section from Page 15 to 17 highlighted by the blue color in the manuscript:

Physical origin of NC effects in antiferroelectric.

In order to explore the origin of the NC effect microscopically, the scanning transmission electron microscope (STEM) is used to record the atom-column information, as displayed in Fig. 6a. The local polarization of the antiferroelectric film PZO can be evaluated by the deviation of the Pb-cation from the geometric center of the four surrounding Zr-site columns. To better compare the Pb displacements, we extract the magnitudes of Pb displacements by averaging the magnitudes of Pb cations along the [110] direction and plot the spatial variation of the displacements along the x-direction in Fig. 6b. The polarization is almost compensated, demonstrating the nearly zero polarization under zero electric field and the antiferroelectricity of the PZO film in a relatively large scale. All the Pb-cation displacements are presented in the polarization vector maps in Fig. 6c, where two dipole arrangement modes are observed (details in Supplementary Note 1 and Fig. S7). The unique polarization configuration is induced by the film strain, which contribute to the tilt and distortion of the oxygen octahedra.^{17,27} Remarkably, the polarization vectors have a fluctuation period of 4 with a couple of two-layer dipole arrangement with different magnitudes or directions. And the polarization in the sublattice has the same direction and changes the directions between the sublattices. Without the applied external electric field, the electric field in the antiferroelectric film arises from the depolarized field. The electric field E in the antiferroelectric PZO film is estimated by the surface density of the polarization charge ρ (Supplementary Fig. S8) which is the negative divergence of the polarization P (see Method section "Estimation of the depolarized field E and free energy G "). The electric field vector E maps are displayed in Fig. 6d, which have different distribution rules with the polarization P .

To build the connection between the polarization and electric field, we plot the y components of the local polarization displacement P_y (yellow line) and the electric field E_y (blue line) along the perpendicular line I (see **c** and **d**) in the upper panel of Fig. 6e. And the free energy G evaluated by $G \approx \int E_y dD_y$ (D represents the displacement field, see Methods section “Estimation of the depolarized field E and free energy G ”) is presented in the lower panel of Fig. 6e. The differential capacitance C is proportional to dP/dE , so the regions where the polarization has the opposite direction to the electric field E exhibit negative capacitance. Meanwhile, the capacitance is related to $\frac{\partial^2 G}{\partial^2 D}$, so the regions where the free energy curve has a negative curvature show negative differential capacitance. It is obvious that in Fig. 6e the light blue shadow regions mark the NC regions. Note that several positions in the perpendicular line I possess negative differential capacitance, and this phenomenon is universal in the whole regions (see Supplementary Fig. 9 for more measurements in different regions). The inhomogeneous polarization distribution in PZO film induces the difference of the polarization magnitudes in adjacent sublattices, further leading to the inhomogeneous distribution of the electric field. The polarization and electric field create the regions where the NC effect emerges. Remarkably, the antiferroelectrics have denser NC local regions than the ferroelectrics, and the NC effect emerges within the domain and not limited by the domain walls. The total capacitance can be regarded as the series and parallel structures of all of the NC regions in three-dimensional space.

Fig. 6 Identifying the local regions of NC effect. **a** Cross-sectional STEM image of the PZO film. **b** Distribution of surface density of polarization charge ρ . **b** Variation of Pb-cation polar displacement averaged the magnitudes along the [110] direction. **c** Polarization vector P mapping extracted from the Pb-cation displacement in **a**. **d** Electric field vector E mapping calculated from the surface density of polarization charge ρ in the same region. **e** Variation in the y component of the polarization displacement P_y (yellow line) and electric field E_y (blue line), and the local energy estimated according to $G \approx \int E_y dD_y$ along the perpendicular line I . The light blue shadows in **e** indicate these regions have the negative differential capacitance where $\partial^2 G / \partial^2 D < 0$ and $dP/dE < 0$.

In addition, I have a few minor remarks.

a) I have a feeling that the observed inhomogeneities are related to the compensation of the internal strain, like ferroelastic domains in strained ferroelectric films. Is this correct?

Author Reply: The Reviewer is correct. The antiferroelectric orderings in pure PZO ceramics have two staggered sublattices with opposite polarization directions and the same magnitude. While the arrangement mode is more complicated in the doped antiferroelectric ceramics. The antiparallel dipole arrangements with different magnitudes or nearly orthogonal arrangements

were observed in doped PZO based ceramics in 2019 (Ref. 27). Zr-rich Pb(Zr,Ti)O₃ ceramics were found to show the magnitude and angle modulation modes with uncompensated polarization, which can be ascribed to the tilt and distortion oxygen octahedral (Ref. 17). In the present case, although the PZO film is pure and not doped, the strain effect from the substrate most likely leads to the tilt and distortion of the oxygen octahedral, accompanied by the polarization configuration. The corresponding description is added to Page 16 Line 5: All of the Pb-cation displacements are presented in the polarization vector maps in Fig. 6c, where two dipole arrangement modes are observed (details in Supplementary Note 1 and Fig. S7). The unique polarization configuration should be induced by the film strain and resultant tilting and distortion of the oxygen octahedral.^{17,27}

b) The explication of the static NC by figure 1 is still misleading. In fact, in Fig.1 the system passes through the “forbidden maximum” at a finite field, hence the related NC has the differential finite-voltage meaning, $C=dU/dQ$. What authors claim is to be observed is the real static $C=U/Q<0$ arising at $U=0$. Appeal to Fig. 1 can only hide the genuine origin of this NC (if the effect is validcorrect).

Author Reply: The Reviewer is correct. The monodomain theory should not be applied to our samples because the present antiferroelectric PZO film possesses multidomain structures (see Supplementary Fig. 1). Thus the original Figure 1 is deleted to avoid possible misleading.

In the non-linear materials, such as the ferroelectrics and antiferroelectrics, the capacitance refers to the differential capacitance, which is defined as

$$C = dQ/dV \text{ or } C = (d^2U/dQ^2)^{-1}, \text{ where } Q, V \text{ and } U \text{ are charge, voltage and}$$

the free energy, respectively. In the free energy curves based on the monodomain model of antiferroelectrics of Fig. 1, the regions around the polarization switching have a negative second derivative of U and Q ($d^2U/dQ^2 < 0$) where U is not equal to zero, meaning that the differential capacitance is negative.

Accordingly, the introduction part is revised. In Page 2 Line 10: This series

circuit can drive the device from the voltage-controlled operational mode into charge-controlled one, which would reverse the polarization and voltage curve from the multi-valued S shape to the single-valued N shape.⁵ The exploration into the mechanism of the static NC effect has developed from the initial monodomain Landau theory to the mechanism based on domains. Luk'yanchuk et al.⁹ constructed an effective model to elaborate the explicit process of the NC effect. Splitting the monodomain state into two-domain ground state and the subsequent motion of the domain wall would modify the energy curve and result in the non-hysteresis curve of charge and voltage with negative slope. In Page 2 Line 4 from the bottom: Reverse domain nucleation and accelerated growth during the switching can induce the switching instability, making the temporal evolution of the charge and voltage follow a negative slope.⁹ In Page 3 Line 4: Besides ferroelectrics, the antiferroelectrics based on Landau switches may have NC effect. For the ideal monodomain antiferroelectrics, the second derivative between the free energy G and the charge Q of the free energy curves could lead to the NC effect^{13,14}. While the realistic structure in antiferroelectrics tends to be multidomain structures due to the competition between the energies¹⁵, and the multidomain structures have been observed in our samples (Supplementary Fig. 1). The sublattice structures in antiferroelectrics further complicate the system and would enrich the physical origin.

c) It would be instructive to plot the depolarization charge distribution $\rho = -\text{div}(P)$, related to Fig. 7.

Author Reply: Thanks for the helpful suggestion. According to the formula $\rho = -\text{div}(P)$, the distribution of the depolarization charge is plotted in the Supplementary Fig. S8a. The depolarization charge fluctuates like wave with positive and negative depolarization charge arranged alternately.

Supplementary Fig. 8 a Distribution of the surface density of the polarization charge ρ .

REVIEWERS' COMMENTS

Reviewer #3 (Remarks to the Author):

Authors satisfactorily explained and confirmed the origin of static NC in the antiferroelectric system via a specific response of the non-uniform domain-clustering state. This gives the new dimension for the NC-related research, showing that the static NC effect is not the property of the only ferroelectrics but can also emerge in more general systems with the nonuniform distribution of the internal bounded (depolarization) charge. This aspect is novel and I recommend the article for publication.

I recommend only to be more precise in the relevant description of the previous achievements.

- Ref. 5 can be displaced for one more sentence ahead and be located at the end of the sentence "...theory to the mechanism based on domains".
- The next two sentences are not really necessary since they discuss the very different case of the ferroelectric two-domain negative capacitor.
- Ref. 11 (Khan et al.) can be also added at the end of the sentence "Reverse domain ... follows a negative slope"

Igor Lukyanchuk

We are grateful to the positive evaluations for our new version by the reviewer: “Authors satisfactorily explained and confirmed the origin of static NC in the antiferroelectric system via a specific response of the non-uniform domain-clustering state. This gives the new dimension for the NC-related research, showing that the static NC effect is not the property of the only ferroelectrics but can also emerge in more general systems with the nonuniform distribution of the internal bounded (depolarization) charge. This aspect is novel and I recommend the article for publication.” We address his suggestions point-by-point below.

Response to the Reviewer:

Comments: Authors satisfactorily explained and confirmed the origin of static NC in the antiferroelectric system via a specific response of the non-uniform domain-clustering state. This gives the new dimension for the NC-related research, showing that the static NC effect is not the property of the only ferroelectrics but can also emerge in more general systems with the nonuniform distribution of the internal bounded (depolarization) charge. This aspect is novel and I recommend the article for publication.

I recommend only to be more precise in the relevant description of the previous achievements.

- Ref. 5 can be displaced for one more sentence ahead and be located at the end of the sentence “...theory to the mechanism based on domains”.
- The next two sentences are not really necessary since they discuss the very different case of the ferroelectric two-domain negative capacitor.
- Ref. 11 (Khan et al.) can be also added at the end of the sentence “Reverse domain ... follows a negative slope”

Igor Lukyanchuk

Author Reply: We appreciate Prof. Igor Lukyanchuk’s positive and kind suggestions. We have modified the related description of the previous achievements in Page 2. The changes include:

1. The Ref. 5 has been moved to the end of the sentence “...theory to the mechanism based on domains”.
2. The sentence describing the ferroelectric two-domain negative capacitor of “Luk’yanchuk et al.⁹ constructed an effective model to elaborate the explicit process of NC effect. Splitting the monodomain state into two-domain ground state and the subsequent motion of the domain wall would modify the energy curve and result in the non-hysteresis curve of charge and voltage with negative slope.” has been deleted.
3. The reference (Khan, A. I. et al. *Nat. Mater.* **14**, 182–186 (2015)) has been added at the end of the sentence “Reverse domain ... follows a negative slope”.